# Peste des Petits Ruminants in Central and Eastern Asia/West Eurasia: Epidemiological Situation and Status of Control and Eradication Activities after the First Phase of the PPR Global Eradication Programme (2017–2021)

**DOI:** 10.3390/ani12162030

**Published:** 2022-08-10

**Authors:** Matteo Legnardi, Eran Raizman, Daniel Beltran-Alcrudo, Giuseppina Cinardi, Timothy Robinson, Laura C. Falzon, Hervé Kapnang Djomgang, Edward Okori, Satya Parida, Felix Njeumi, Camilla T. O. Benfield

**Affiliations:** 1Food and Agriculture Organization of the United Nations (FAO), Viale delle Terme di Caracalla, 00153 Rome, Italy; 2Regional Office for Europe and Central Asia, Food and Agriculture Organization of the United Nations (FAO), 1068 Budapest, Hungary

**Keywords:** peste des petits ruminants, small ruminant morbillivirus, disease eradication, sheep, goats, small ruminant, wildlife, Asia, Economic Cooperation Organization, PPR GEP

## Abstract

**Simple Summary:**

Peste des petits ruminants (PPR) is a highly contagious viral disease of domestic and wild small ruminants. The disease is endemic to large parts of Africa, the Middle East and Asia and causes severe socioeconomic losses, especially in developing countries reliant on small ruminant value chains. Currently, PPR is the only animal disease targeted by the Global Eradication Programme (PPR GEP), which aims to eradicate the disease by 2030. Following the end of the first five-year phase of the PPR GEP, the goal of this review is to provide an update on the status of the eradication progress in one of the nine regions targeted for coordinated action in the PPR Global Control and Eradication Strategy, denominated Central Asia/West Eurasia. In addition to the original nine countries, regional meetings and activities have involved four additional countries based on shared epidemiological features, which are also reviewed here. The considered area spans from Eastern Europe to East Asia and features remarkable variability in terms of both PPR presence and enacted control efforts. The achievements and constraints encountered at regional and national levels are discussed, thus providing useful data for tailoring the next steps of the eradication programme to the peculiarities of the region.

**Abstract:**

Peste des petits ruminants (PPR) is a highly contagious infectious disease of small ruminants caused by peste des petits ruminants virus (PPRV). PPR poses a significant threat to sheep and goat systems in over 65 endemic countries across Africa, the Middle East and Asia. It is also responsible for devastating outbreaks in susceptible wildlife, threatening biodiversity. For these reasons, PPR is the target of the Global Eradication Programme (PPR GEP), launched in 2016, which is aimed at eradicating the disease by 2030. The end of the first five-year phase of the PPR GEP (2017–2021) provides an ideal opportunity to assess the status of the stepwise control and eradication process. This review analyses 13 countries belonging to Eastern Europe, Transcaucasia, and Central and East Asia. Substantial heterogeneity is apparent in terms of PPR presence and control strategies implemented by different countries. Within this region, one country is officially recognised as PPR-free, seven countries have never reported PPR, and two have had no outbreaks in the last five years. Therefore, there is real potential for countries in this region to move forward in a coordinated manner to secure official PPR freedom status and thus reap the trade and socioeconomic benefits of PPR eradication.

## 1. Introduction

Peste des petits ruminants (PPR) is a severe and highly contagious viral disease of small ruminants (SR) whose economic impact has been estimated at USD 1.4–2.1 billion annually [1]. First described in 1942 in Côte d’Ivoire [2], in the last two decades, PPR has been detected in over 65 countries across Africa, Asia, the Middle East and Europe [3], threatening more than 80% of the global sheep and goat population [4]. In addition, PPR has been recognised as responsible for clinical outbreaks in multiple wild ruminant species [5,6,7,8,9], including already endangered species, thus posing a demonstrated threat to biodiversity [7,10,11]. Evidence of infection was also found in other ruminants, such as camels, which may develop clinical signs, as well as cattle and buffaloes, which become subclinically infected but seroconvert [12]. Nonetheless, the determinants of disease expression in wild and atypical hosts, and the role of these hosts in PPRV circulation and maintenance, are not well understood, and may well vary between different ecosystems [11,13,14].

The causative agent of PPR is peste des petits ruminants virus (PPRV), which belongs to the family *Paramyxoviridae* and the genus *Morbillivirus.* The same genus includes other burdensome veterinary pathogens affecting domestic and wild mammals, including canine distemper virus, phocine distemper virus and rinderpest virus. The latter is responsible for rinderpest, a disease of predominantly large ruminants, which shares many features with PPR and is the only animal disease to have been eradicated, as declared in 2011 [15]. PPRV is an enveloped virus with a single-stranded, negative-sense RNA genome. Its genetic variability is comparable to that of most RNA viruses, with a substitution rate in the order of 10^−3^/10^−4^ substitutions per site per year [10,16,17]. There are four different PPRV lineages (I–IV), which can be discriminated based on a portion of the N gene [18]. Historically, lineages I–II have been mostly detected in West Africa, while lineage III has circulated in the Middle East and East Africa [16,19]. After its likely origin in West Africa [10,16], lineage IV spread eastward and became the predominant lineage across Asia; it then re-emerged in Africa, where it now seems to be the prevalent lineage as well [19].

Despite these genetic differences, all lineages belong to a single serotype, and the currently available live attenuated vaccines confer a long-lasting and effective protective immunity against all of them [20,21]. The availability of effective preventive and control tools, along with the lessons learned during the rinderpest eradication campaign, laid the foundation for the PPR Global Eradication Programme (PPR GEP). This programme was launched in 2016 by the Food and Agriculture Organization of the United Nations (FAO) and the World Organisation for Animal Health (WOAH, formerly OIE), with the aim of eradicating the disease by 2030. PPR eradication will help in promoting the contribution of small ruminant systems to food security and economic growth. The recent conclusion of the first five-year phase of the PPR GEP (2017–2021) represents the ideal moment to evaluate the progress made thus far by countries and regions. 

To facilitate the exchange of information with national veterinary services, harmonise surveillance and control efforts, and promote coordination between neighbouring countries, the PPR Global Control and Eradication Strategy (GCES) established nine different regions as targets for Regional Roadmap meetings, overseen by Regional Advisory Groups (RAGs). These regions were defined primarily according to the distribution of FAO and WOAH regions and subregions and to the existence of relevant regional economic communities. An epizone approach, involving the grouping of countries with similar epidemiology, was also envisioned to promote the cooperation of neighbouring countries that share epidemiological features but that may have been included in different Roadmap regions [1].

The aim of this review is to provide an overview of the PPR epidemiological situation and the status of the eradication campaign with a focus on one of these regions, denominated Central Asia/West Eurasia. Originally, this region, as defined in the PPR GCES, included nine countries, namely, Armenia, Azerbaijan, Georgia, Kazakhstan, Kyrgyzstan, Tajikistan, Turkey, Turkmenistan and Uzbekistan [1]. The region largely overlaps with the area of the Economic Cooperation Organization (ECO), whose membership includes Azerbaijan, the Islamic Republic of Iran, Kazakhstan, Kyrgyzstan, Tajikistan, Turkey, Turkmenistan and Uzbekistan. ECO, established in 1985, is an intergovernmental organisation aimed at promoting the economic development of its member states [22], and its Secretariat has been involved as an active partner in the organisation of periodic Roadmap meetings held in the region and in the formulation of a regional eradication strategy. Despite being ECO members, Afghanistan and Pakistan were not assessed in this work, as they are also part of the South Asian Association for Regional Cooperation (SAARC) and were included in the South Asia region [1]. On the other hand, based on the above-mentioned epizone approach, four additional countries (China, the Islamic Republic of Iran, Mongolia and the Russian Federation) have been involved in activities conducted in the region, particularly the latest Roadmap meeting held in October 2021, and were also considered in this review.

The current PPR epidemiological situation, both at regional and at national levels, is provided using the stepwise approach proposed in the PPR GCES to track each country’s progress towards PPR freedom status [1]. This stepwise pathway has four stages: stage 1 (assessment stage), stage 2 (control stage), stage 3 (eradication stage) and stage 4 (post-eradication stage). In stage 4, vaccination should be suspended, and appropriate evidence should be collected proving that PPRV is no longer circulating in the country, aiming to apply to WOAH for official PPR-free status. At each of these stages, the activities to be undertaken are divided into five technical elements: the diagnostic system, the surveillance system, the prevention and control system, the legal framework for PPR prevention and control, and stakeholders’ involvement in PPR prevention and control activities [1]. The implementation of this pathway enables the collection of standardised information from countries and the individuation of the main achievements and constraints in control and eradication activities undertaken in the region, as presented below. 

## 2. Methodology

The information presented here was obtained from countries’ contributions to Regional Roadmap meetings organised during the first phase of the PPR GEP. These were held in Almaty (Kazakhstan) in February 2016, Dushanbe (Tajikistan) in February 2017, Tashkent (Uzbekistan) in August 2019, and virtually in October 2021. During these meetings, countries’ representatives were asked to provide updates on the PPR epidemiological situation and on the relevant activities conducted at the national level using templates provided by the Joint FAO-WOAH PPR Secretariat. Participants were also involved in plenary discussions to promote the exchange of information and cooperation at the regional level. In addition, countries were asked to conduct a periodic self-assessment using the provided PPR Monitoring and Assessment Tool (PMAT). Armenia, Azerbaijan, China, Georgia, the Islamic Republic of Iran and Turkey all submitted updated PMAT reports on the occasion of the last Roadmap meeting, and their responses were considered. Lastly, the PPR Central Asia Regional Strategy, formulated in 2017, was also taken into account to retrieve information about the history of PPR in the region and the different strategies adopted by single countries and coordinately. 

The data reported by countries were integrated with relevant information retrieved from the scientific literature. The OIE World Animal Health Information System (WAHIS) was consulted for each country’s official disease status [3], while official figures on small ruminant (SR) population numbers (updated to 2020) were retrieved from FAOSTAT [23]. The data regarding SR population density at the subnational level are from the fourth iteration of the Gridded Livestock of the World (GLW 4) database (unpublished), generated as described by Gilbert et al. [24]. Information about the date and location of PPR outbreaks that occurred in the region between 2012 and 2022 were obtained from EMPRES-i [25], from EU-ADIS (for Turkey) [26] and through the Iran Veterinary Organization (for the Islamic Republic of Iran) [27]. Maps were prepared using QGIS 3.12 [28].

## 3. PPR Situation at Regional Level

The region under consideration spans from Eastern Europe to Eastern Asia, also including countries from Western and Central Asia, and covers an area of about 33,878,606 km^2^. In 2020, the total SR population in the region amounted to more than 380 million sheep and 203 million goats, representing 30% and 18% of their respective global populations [23]. Each country’s SR population is presented in Table 1.

PPR presence and the implemented surveillance and control measures vary considerably across the region depending on the production system. SR systems include nomadic and semi-nomadic forms of pastoralism, which are still widely practised in most of Central Asia, South Caucasus [29], Mongolia [30] and the Islamic Republic of Iran [31], and sedentary household farming, which is prevalent in large parts of Turkey [32], China [33] and the Russian Federation [34]. Most countries have a mixture of pastoralism, with seasonal animal movements and more sedentary systems in other parts of the country. Figure 1 shows the domestic SR population distribution in the region, along with the location of PPR outbreaks that occurred between 2012 and 2022.

PPR is currently reported in four of the countries under consideration, namely, the Islamic Republic of Iran, Turkey, China and Mongolia; in Georgia and Tajikistan, it has not been reported since 2016 and 2013, respectively, while in the seven remaining countries, the disease has never been officially reported (Figure 2).

The fact that PPR detections are limited to the southernmost countries may be attributed to several reasons. Based on ecological niche modelling, all of the infected countries represent highly suitable niches for PPRV, with maximum and minimum temperatures and SR density being the most relevant factors [35]. However, other historically free countries in Central Asia and Caucasus feature the same risk factors. It therefore seems clear that sharing borders with endemic countries located outside of the considered area likely plays a role in PPR introduction. For instance, a predictive analysis study recently suggested the existence of several epidemiological corridors between China and neighbouring endemic countries, such as India, Nepal, Bhutan and Bangladesh, and proposed transhumance and the presence of susceptible wildlife as factors likely contributing to PPR transmission and maintenance in the Trans-Himalayan region [36]. Using a similar least-cost path (LCP) modelling approach for transboundary PPR movement, epidemiological linkages were predicted between China and Pakistan, India, Kazakhstan and Tajikistan [37].

Lineage IV seems the only lineage to be circulating in the whole region, as supported by molecular studies conducted in the Islamic Republic of Iran [38,39], Kazakhstan [40], Tajikistan [41,42], Georgia [43], Mongolia [10,44], Turkey [45,46,47] and China [48,49,50,51,52]. However, some PPRV strains from the region belong to distinct subclades of lineage IV, allowing the inference of separate patterns of local circulation and transboundary spread. While some of these findings are fairly easy to explain in light of geographic proximity or trade relations, some others are more difficult to rationalise. For example, the strains detected in Georgia were found to be closer to Northern and Eastern African sequences than the ones found in neighbouring countries [43,53]. This suggests that some of the epidemiological linkages supporting PPRV circulation in the region may still be poorly understood and long-range. 

The variability in terms of PPR presence is reflected by differences in the surveillance and control efforts put in place by different countries, which in turn result in countries being at different stages of the PPR GCES stepwise approach, ranging from stage 1 to official PPR freedom status (post stage 4) (Figure 3).

Marked heterogeneity is also observed in the budget devoted by each country to activities related to PPR (Table 2). In 2020, the last year for which data were available, the total reported PPR budget across all 13 countries was USD 40.8 million, with China accounting for USD 23 million of this sum.

Of the countries under consideration, Azerbaijan, Mongolia and two laboratories in Turkey have participated since 2016 in the PPRV Interlaboratory Tests for Molecular and Serological Diagnostics conducted by the Animal Production and Health Laboratory of the Joint FAO/IAEA Centre of Nuclear Techniques in Food and Agriculture. However, additional countries also report undertaking proficiency testing, presumably with other organisations. The Central Asia Animal Health Network (CAAHN) was established in 2019 under the lead of FAO and WOAH and helps promote the coordination of animal health initiatives in Central Asia, so far prioritizing neglected zoonoses. However, relevant assessments of the surveillance and evaluation capacities have been conducted under CAAHN using FAO’s Surveillance Evaluation Tool (SET) and the Laboratory Mapping Tool (LMT).

To harmonise the efforts towards the eradication of the disease, a PPR Regional Strategy for Central Asia was formulated in 2017 with support from the ECO Secretariat. This strategy, however, has never been endorsed by member countries. Another initiative that could support PPR eradication is the establishment of the Veterinary Commission of ECO member states (ECO-VECO), which was put into action in 2018 by the ECO Heads of Veterinary Organizations, although it has not yet been approved by all members. ECO-VECO is aimed at improving national veterinary services and promoting the trade of animal products within member countries via the organisation of joint training courses and research programmes, the development of international standards for livestock identification and traceability, and the harmonisation of disease control activities. 

Despite the efforts to harmonise the approach towards the eradication of the disease, many differences persist from one country to another. To better understand the reasons behind these differences, detailed information about each country’s epidemiological status and implemented activities are provided below.

## 4. PPR Situation at National Level

### 4.1. Armenia

PPR has never been reported in the Republic of Armenia. According to the data presented at the 2021 Roadmap meeting, risk-based targeted surveillance activities were performed in the country by serological means in 2019–2020. In these two years, 3280 sera were collected in the regions of Syunik, Gegharkunik and Armavir, where most of the national SR population is farmed, and no evidence was found of PPRV circulation. The diagnostic capacity available at the Laboratory for Especially Dangerous Pathogens of the Republican Veterinary-Sanitary and Phytosanitary Centre for Laboratories Services (SNCO) includes reverse transcription polymerase chain reaction (RT-PCR) in addition to PPRV antigen and antibody detection methods. However, several constraints have been reported regarding diagnostics and surveillance activities, such as the lack of genetic sequencing capacity, routine surveillance activities in wildlife and data on PPR’s socioeconomic impact. As for prevention and control, a targeted vaccination campaign was conducted in 2016–2017 in buffer zones neighbouring Georgia, Turkey and the Islamic Republic of Iran. In detail, 245,112 and 241,238 heads were vaccinated in 2016 and 2017, respectively, but no information was collected on post-vaccination seroconversion rates. Appropriate biosecurity measures are also reportedly taken at borders, banning the import of SR and SR products from endemic countries, but prevention efforts are reported to be hampered by the lack of an animal identification system and by the PPR National Strategic Plan (NSP) not yet being officially endorsed by the government. The budget for PPR-related activities comes primarily from national funding, which ranged from USD 2 million in 2017 to USD 2.9 million in 2020. Additional funds were provided in 2019–2020 by the Defense Threat Reduction Agency (DTRA), USA, in the context of a project to detect the origin and distribution of PPR in both Armenia and Georgia (‘Comprehensive Research for Detecting Origin and Distribution of Peste des Petits Ruminants (PPR) in Georgia and Armenia’, reference number BAA-PPR #HDTRA11810053). Based on self-assessment, as of 2021, Armenia is in the fourth stage of the stepwise approach, and application for freedom is planned for 2022.

### 4.2. Azerbaijan

PPR has never been reported in the Republic of Azerbaijan, and vaccination has never been conducted. Surveillance activities are carried out at the Central Veterinary Laboratory of the Azerbaijan Food Safety Institute and rely on different diagnostic methods, including both antigen (Ag) and antibody (Ab) ELISAs and PCR assays. The most recent and comprehensive data were obtained in 2020, when, with the support of FAO, a serosurvey was conducted by sampling 10,032 adult female SR and cattle from randomly selected herds located in 28 different districts, revealing a 2.03% seroprevalence. The 206 seropositive samples came from 142 farms, where a follow-up investigation was conducted on a total of 3577 animals, which all gave negative antigen test results. The reported seropositivity rate could potentially be explained by the non-specificity of the diagnostic test, especially in the case of cattle sera, or by the sampling of imported animals. The legal framework reportedly includes laws allowing for SR identification and traceability, restrictions on animal movements, the enactment of biosecurity measures and compensation for farmers in case of culling for eradication purposes. In light of this, Azerbaijan is currently in the fourth stage of the stepwise approach and is preparing an application for official PPR-free status with the support of FAO.

### 4.3. China

The People’s Republic of China is home to the largest population of sheep in the world, as well as the second largest goat population (Table 1). PPR was first reported in China in July–November 2007, when a series of outbreaks were observed in domestic SR in southwestern Tibet [54]. Following the implementation of strict control measures, no evidence of PPRV circulation had been found in Tibet by 2010. However, the disease re-emerged in Xinjiang province in late 2013, but this time, it quickly spread to most of the country, involving more than 22 provinces, autonomous regions and municipalities in 2014 alone [55]. The swift spread of PPR has been attributed to the transport of breeding stock of infected sheep and goats from Xinjiang to markets located in different parts of the country in the spring of 2014 [55,56]. At the time, most of the Chinese SR population was not immunised against PPR, since compulsory vaccination was adopted only in Tibet and Xinjiang [55]. Based on phylogenetic analyses, the Tibetan strains from 2007–2008 were more closely related to viruses found in India in 2014–2016 than to viruses circulating in China since 2013 [10,51,57], suggesting that the two waves may have originated from separate PPRV introductions. Fewer but regular outbreaks were reported in the years following 2014, and, as of 2021, the occurrence of PPR in domestic herds appears to be mostly limited to Xinjiang. In addition, PPRV infections in Chinese wildlife have been reported for many years. During the Tibetan outbreak, several bharals (*Pseudois nayaur*) were found to be positive by RT-PCR [8], and molecular detection was also reported in ibexes (*Capra ibex sibirica*), argali sheep (*Ovis ammon*) and goitered gazelles (*Gazella subgutturosa*) during the 2013–2014 outbreak [56] and in a Przewalski’s gazelle (*Procapra przewalskii*) found dead in 2018 in Gansu province [58].

Currently, a government-funded National Surveillance Project is in place to monitor PPR circulation in both domestic and wild populations. The diagnostic tests employed for PPR diagnosis at the China Animal Health and Epidemiology Center (CAHEC), a WOAH Reference Laboratory and FAO reference centre for PPR, include ELISA methods, RT-PCR, quantitative reverse transcription PCR (qRT-PCR) and both Sanger and Next-Generation Sequencing. Since 2014, a countrywide National Compulsory Immunisation Project has also been implemented. Vaccination coverage and seroconversion are assessed annually: according to the official figures for 2020, 98.9% of the domestic SR population was vaccinated with 92.7% seroconversion, comparable to rates reported in previous years. Other implemented measures include an animal identification system and an Emergency Plan to be followed in case of PPR outbreaks. Despite the efforts already undertaken to control SR movements within the country, many difficulties reportedly remain in implementing it effectively in such a large country. According to the self-evaluation by PMAT performed in 2021, China is in the third stage of the PPR GEP stepwise approach.

### 4.4. Georgia

The first and only officially notified PPR outbreak in Georgia occurred in early 2016 in sheep farms located near Tbilisi [43]. Subsequent molecular investigations did not enable the inference of the origin of the detected lineage IV strains, which were closer to Northern and Eastern African sequences than the ones found in neighbouring countries, possibly suggesting an unknown epidemiological linkage [43,53]. Since the first notification, surveillance efforts have included both passive activities, with 76 suspected cases testing negative by PCR, and an ongoing active serosurvey funded by the USA Defense Threat Reduction Agency (DTRA), under which 3200 sera were tested in 2019–2020. Surveillance is not conducted in wildlife. A vaccination campaign in newborn SR has been undertaken since 2016, with plans to vaccinate in 2022, targeting newborns and border areas, but not vaccinate in 2023. The vaccination campaign benefited from the donation of 400,000 vaccine doses by FAO at the end of 2020 and 500,000 in 2022. The most recent annual coverage data, presented at the 2021 Roadmap meeting, are as follows: 2016: 1,650,974 SR vaccinated, 81% seroconversion (based on 373 samples);2017: 341,461 SR vaccinated, 74% seroconversion (based on 499 samples);2018: 339,655 SR vaccinated, 30% seroconversion (based on 100 samples);2019: 307,809 SR vaccinated, 64% seroconversion (based on 197 samples);2020: 279,832 SR vaccinated, 38% seroconversion (based on 100 samples);2021: 643,088 SR vaccinated, 84% seroconversion (based on 100 samples).

In addition, an emergency vaccine stock is maintained in case of PPR outbreaks. An SR identification system is in the process of being implemented, and veterinary surveillance points have been instituted along migration routes. According to a participatory study conducted by Chenais et al. [59], Georgian SR farmers were familiar with the disease, supporting the effectiveness of the awareness activities reportedly conducted during the vaccination and serosurvey campaigns. No unreported PPR outbreaks were suspected based on the study results. As of 2022, Georgia is placed in the third stage of the stepwise approach.

### 4.5. Islamic Republic of Iran

In 2019, the Islamic Republic of Iran had the fifth largest sheep population in the world, totalling more than 41 million heads, and a goat population of 15 million [23]. It is therefore not surprising that, ever since the first documented outbreak in 1995, PPR has represented a severe economic and health threat to the country [60]. The issue is not limited to domestic SR: a significant concern is also caused by the impact of PPR on Iranian wildlife, including vulnerable species of wild sheep (*Ovis orientalis*) and goats (*Capra aegragus*). Marashi et al. [5] identified PPR as the cause of multiple outbreaks that occurred in different national parks between 2014 and 2016, resulting in more than 1000 deaths of wild sheep and goats. However, PPR involvement in outbreaks in wildlife has been suspected since at least 2000 [5]. In addition, a PPR clinical outbreak was also documented in a camel herd in 2013 and confirmed by serology and RT-PCR [61]. 

The national PPR control and eradication strategy is based on five pivots, namely, diagnosis, epidemiological surveillance, control measures, training and awareness, and monitoring and evaluation. Diagnostic activities rely on virus isolation, Ab and Ag ELISAs, RT-PCR and sequencing, conducted at the national reference laboratory and in three subregional laboratories. Passive surveillance is conducted in domestic species, and active surveillance is underway in wildlife. Vaccination of domestic SR has been carried out since 2008, and a live attenuated vaccine (based on the Nigeria 75/1 strain) has been produced locally since 2015. The subsequent increase in vaccination coverage reportedly led to a decline in disease outbreaks in both domestic and wild populations. Nonetheless, no identification system is in place for small ruminants, and the effectiveness of vaccination campaigns is hampered by difficulties in controlling the movements of nomadic herds and by the lack of post-vaccination evaluation. Close contacts between domestic and wild animals are also reported as a problem for PPR control. Currently, the Islamic Republic of Iran is in the second stage of the stepwise approach.

### 4.6. Kazakhstan

Officially, PPR has never been reported in the Republic of Kazakhstan. Analyses are conducted at the Republican State Enterprise (RSE) National Reference Centre for Veterinary Medicine using Ag and Ab ELISA assays and qRT-PCR. The NSP, which was developed in 2017 and endorsed in 2018, is based on a division between high- and low-risk zones. The high-risk zone comprises the southern regions of East Kazakhstan, Zhambyl, Almaty, Kyzylorda and Turkestan, where 60% of the total SR population is located, while the rest of the country comprises the low-risk zone. Vaccination is carried out only in the high-risk zone, aiming to prevent PPR incursion from bordering countries [40]. From 2017 to 2021, between 12% and 42% of the total SR population was vaccinated. However, post-vaccination evaluation (PVE) has not been conducted, while serosurveillance is conducted twice a year by simple random sampling, both in the low-risk zone (in all SR) and in the high-risk one (only in unvaccinated young animals). Based on this information, Kazakhstan has self-assessed the low-risk zone to be in the fourth stage of the PPR GCES stepwise approach and the high-risk zone to be in the third stage.

Nonetheless, it should be noted that the results of scientific studies suggest that PPRV may have been circulating in the country undetected. Lurdervold et al. [62] first reported the detection of anti-PPRV antibodies in a small number of sheep, goats and cattle in central Kazakhstan in 1997–1998. Over 15 years later, another study found PPRV to be responsible for clinical outbreaks in three separate farms in late 2014 in the Zhambyl region [40]. Based on partial N gene sequencing, the identified strains showed high similarity to Chinese strains from 2013/2014, suggesting PPR transboundary spread between the two countries. The three outbreaks did not have any obvious epidemiological linkage, suggesting that PPRV may have been persistently present at a subclinical level despite vaccination efforts [40]. This finding is even more worrying when considering that central Kazakhstan is home to the largest global population of saiga antelope [63], which suffered a population crash in Mongolia due to PPR in 2016–2017. However, no PPR seropositive saiga antelopes were found in a serological investigation conducted in Kazakhstan between 2012 and 2014 [63].

### 4.7. Kyrgyzstan

PPR has never been reported in the Kyrgyz Republic, which is currently in the third stage of the stepwise approach. Nonetheless, PPR control is part of a permanent Control and Prevention Strategic Plan, mostly funded through the national budget, aiming to demonstrate freedom from the disease and viral circulation.

Both active and passive surveillance activities are conducted by the Centers for Veterinary Diagnosis and Expertise of the Northern and Southern regions of the State Inspectorate on Veterinary and Phytosanitary Safety (SIVPS) using Ag and Ab ELISAs. Yapici et al. [64] reported a 35% PPRV seroprevalence in 655 Jaydara sheep aged 1–2 years from four different regions in 2014. However, it is unclear whether the animals’ vaccination status was considered.

Mass vaccination began in 2012 in the southern regions and in buffer zones established at borders with China, Kazakhstan, Tajikistan and Uzbekistan. Since 2016, only young animals born in the respective year have been vaccinated. From 2017 and 2021, the yearly vaccination coverage was between 51% and 58% of the national SR population. In 2019, it was reported that the average seroconversion rate, measured 45–60 days after vaccination, was 88.3%, and it was above 75% in all but two regions. Kyrgyzstan received FAO-supplied vaccines in January 2021 (1 million doses) and March 2022 (1.5 million doses).

### 4.8. Mongolia

PPR presence was first reported in the country in August 2016, when several outbreaks were observed in western Mongolia (Khovd region), possibly as a result of uncontrolled animal movements across the border with China [10,44]. After a large-scale vaccination campaign was conducted in the region from 10 to 30 October 2016, involving more than 4.6 million sheep and 5.8 million goats from five different provinces, this initial outbreak was believed to have been contained. However, PPRV emerged in susceptible wildlife populations, leading to mortality events in saiga antelopes (*Saiga tatarica mongolica*), Siberian ibex (*Capra sibirica*), Argali (*Ovis ammon*) and goitered gazelles (*Gazella subgutturosa*) [7,10]. In particular, the mass die-offs observed in saiga antelopes in 2016–2017, which caused the loss of around 80% of a population that was already critically endangered [7,11], are considered the most important instance of the potential impact of PPR on wildlife. Full genome phylogenetic reconstruction was used to infer the times of most recent common ancestor (TMRCA) viruses and indicated that PPRV was circulating undetected in Mongolia for at least 6 months before the first reported outbreak in August 2016 and that wildlife was likely infected before livestock vaccination began [10]. Since 2018, sporadic outbreaks of the disease have occurred in livestock in the central and south-eastern parts of Mongolia, with the last one being reported in April 2022 [65]. However, the current epidemiological scenario in the country is unclear, and it is difficult to establish whether these events originated from endemic PPRV circulation or from cross-border reintroductions. A PPR eradication strategy is in the process of being updated, but reportedly, field activities lack a long-term strategic approach and are also hampered by resource and budget constraints. According to the assessment made in 2017, Mongolia was in the first stage of the stepwise approach.

### 4.9. Russian Federation

PPR has never been reported in the Russian Federation, which is the only country in the region with WOAH official PPR-free status, gained in 2020. To ensure PPRV absence, passive surveillance is reportedly carried out by the relevant stakeholders at all stages of production, aiming to detect suspected clinical cases for further investigation by virological means. In addition, according to a presentation delivered at the 2019 Roadmap meeting, routine serological surveillance activities are conducted in both farmed and wild susceptible animals in high-risk areas bordering China and Mongolia. In particular, 32,000, 35,333 and 28,500 sera were tested in 2017, 2018 and part of 2019, respectively, but no seropositive cases were reported. The competent body for PPR diagnosis is the Federal Centre for Animal Health (FGBI ARRIAH), whose diagnostic capacity includes Ab ELISA, a validated in-house microneutralisation assay and qRT-PCR. Vaccination is legally banned, but a PPR vaccine stockpile is maintained for emergency vaccination in case of disease outbreaks. Russian-produced PPR vaccines based on the Nigeria 75/1 strain are used in several countries in the region, including Kazakhstan, Tajikistan and Turkmenistan.

### 4.10. Tajikistan

PPR was first reported in the Republic of Tajikistan in 2004 [41]. Before that, local outbreaks with features consistent with PPR were attributed to pasteurellosis [41]. Following the first detection and molecular characterisation, the disease was found in more than half the country, particularly in the southern regions. However, the last officially reported clinical case occurred in 2013 [66]. During a serosurvey conducted in 2014 on 1886 SR sampled across the country, PPR-positive sera were found in 54 of the 65 districts of Tajikistan; however, since vaccination was being carried out, it was not possible to establish whether this was vaccinal or natural immunity. In the following years, only passive surveillance has been carried out; vaccination activities have been conducted in limited parts of the country and are not followed by PVE. In fact, the declared vaccination coverage in 2018 was 5% nationwide and 9% in the Khatlon region, the most SR-populated area and one of the regions that historically faced more problems with PPR. Other documented risk factors for PPR entry and spread in Tajikistan are represented by cross-border contraband trade of SR (especially through the southern borders with Afghanistan, where the disease is endemic) and by the relevance of transhumant pastoralism within the nation.

PPR diagnosis is performed using ELISA, RT-PCR and virus isolation at the National Diagnostic Centre for Food Security. As of 2019, Tajikistan was placed in the first stage of the stepwise approach and had yet to endorse an NSP. Despite PPR not being reported since 2013, Amirbekov et al. [42] described the detection of PPRV strains belonging to lineage IV at least until 2017, along with a remarkable seroprevalence across all administrative districts, which is likely not attributable solely to vaccination. This may suggest that PPR remains endemic in the country.

### 4.11. Turkey

The first detection of PPRV in Turkey occurred in 1999 in Eastern Anatolia [67]. Over the years, PPR has been the object of several studies, which reported its occurrence in different parts of the country with variable degrees of serological and virological prevalence [45,46,68,69]. Evidence of PPRV infection was also found in wild goitered gazelles (*Gazella subgutturosa subgutturosa*), although the mortality observed in this species seems lower than in other susceptible wild hosts [70]. According to passive surveillance activities conducted throughout the country, the number of reported outbreaks peaked in 2017 (100), 2018 (129) and 2019 (109) but was lower in 2020 (53). Vaccination has been performed since 1999, first with a rinderpest vaccine and, since 2002, with a locally produced live attenuated PPR vaccine (Nigeria 75/1 strain). Currently, vaccination is carried out annually for all newborns and unvaccinated adults. Since 2010, more than 13 million SR have reportedly been vaccinated on average each year. Seroconversion rates were 93% in 2018 and 84% in 2020. However, vaccination ceased in Thrace in March 2021. This measure, along with strict regulation of SR movements from Anatolia, has been taken due to the imminent threat that the Thrace region poses for potential PPR introduction to neighbouring European countries [71]. Turkey is aiming to apply for zonal freedom status for Thrace in 2023. According to both active risk-based surveillance activities and independent research projects [46] conducted in the region in recent years, PPR prevalence indeed appears to be lower in Thrace than in Anatolia. Nonetheless, the relevance of proper PPR monitoring and control in this region is demonstrated by the fact that the first PPR outbreaks ever reported in Europe occurred in 2018 in Bulgaria, in areas near the border with Thrace [43], which were promptly contained. PPR laboratory analyses are conducted at the Veterinary Control Central Research Institute; diagnostic capacity includes ELISA tests, virus isolation, RT-PCR, qRT-PCR and sequencing. In terms of socioeconomic impact assessment, a Veterinary Strategy Document of 2016, supported by the EU, concluded that it was beneficial to eradicate the disease in Turkey. According to PMAT evaluation, as of 2021, Turkey is in the third stage of the stepwise approach.

### 4.12. Turkmenistan

PPR has never been reported officially in Turkmenistan. To prevent the introduction of the disease, strengthened surveillance activities and vaccination are performed in buffer zones established along the borders with Iran, Afghanistan and Uzbekistan. Based on the information presented by national representatives during the 2021 Roadmap meeting, surveillance activities consist of passive clinical observation, followed by serological testing of suspected animals. As for vaccination, in recent years the number of vaccinated SR was 842,620 in 2017, 91,341 in 2018, 1,067,814 in 2019 and 797,001 in 2020. No data are available regarding post-vaccinal seroconversion. In 2021, Turkmenistan was in the second stage of the PPR GCES stepwise approach.

### 4.13. Uzbekistan

PPR has never been reported in the Republic of Uzbekistan, and vaccination is not performed. However, Uzbekistan is under constant threat of disease introduction from neighbouring countries, particularly Afghanistan. The measures in place to avoid this possibility include active surveillance activities in buffer zones at the borders with Afghanistan and Tajikistan and the strict control of the import–export of SR. However, the reported number of analysed samples appears to be limited (several hundred per year), and 10–15% of the samples arriving at the laboratory are reported to be unsuitable for serology. Based on self-evaluation, Uzbekistan is currently placed in the second stage of the stepwise approach.

## 5. Discussion

Despite the differences in epidemiological status and PPR risk among different countries of the considered region, several patterns can be identified in both achievements and limitations in terms of diagnostics, surveillance and control activities. Diagnostic capacity appears to be adequate, and proficiency tests are reportedly routinely conducted in most of the countries, but some limitations seem to persist. In particular, limited availability of diagnostic kits has been reported by representatives of multiple countries, including Armenia, Kazakhstan, Tajikistan and China (specifically, in China’s case, following the 2018 African Swine Fever outbreak, which diverted resources), and thought to also be an issue for Uzbekistan and Turkmenistan. The need for specific training to further develop diagnostic capability, both at laboratory and field levels, has also been raised by the representatives of Georgia, Tajikistan and Uzbekistan. 

Some notable shortcomings are also observed in terms of surveillance, possibly the most crucial technical element since it lays the foundation for any possible intervention strategy. The surveillance activities carried out by some countries seem insufficient to capture the local PPR epidemiological features, leading, in some cases, to discrepancies between official reports and independent scientific studies. Another issue related to surveillance is the different attention paid to wildlife among different countries. Either through sporadic or structured monitoring activities, PPRV has been found to circulate in susceptible wild populations in China [8,56,58], the Islamic Republic of Iran [5], Turkey [72] and Mongolia [7]. In particular, the PPR outbreaks that occurred in the saiga antelope in Mongolia clearly demonstrate the importance of setting up a surveillance system capable of detecting PPRV circulation in wild hosts, both for protecting wildlife and for implementing the PPR GCES. 

Vaccination activities are conducted in all countries where PPR is currently present or has been reported in the past, as well as in several countries where PPR is not present but there is the risk of introduction. In particular, a zonal approach, with vaccines being administered only in border regions or in high-risk areas, is adopted by several countries (Armenia (until 2018), Kazakhstan, Kyrgyzstan, Turkey and Turkmenistan), demonstrating the attention given by countries to prevent PPR cross-border entry or further spread to PPR-free areas. Nonetheless, conducting a national vaccination campaign per se is not sufficient to achieve PPR eradication: according to the PPR GEP, vaccination efforts should be coordinated and aimed at achieving a high coverage over a limited period of time, rather than maintaining a constant but lower coverage. Countries where PPR is not detected should eventually cease vaccination, confirm their freedom from the disease by conducting active surveillance for the following 24 months, and comply with the other provisions of the OIE/WOAH Terrestrial Animal Health Code in order to apply for official PPR-free status [73,74]. PVE should also be routinely performed to evaluate the efficacy of the whole vaccination process in providing protective immunity to the disease; based on reports by country representatives, such activities are not always carried out, potentially diminishing the effectiveness of vaccination efforts and hampering the evaluation of progress towards PPR eradication. Another possible limitation, noted during the Roadmap meeting held in 2019, is the absence of laboratories for independent quality control of PPR vaccines in Asia. Other requirements related to prevention and control that are not met homogeneously throughout the considered countries are the implementation of a small ruminant identification system and animal movement traceability (both at national and cross-border levels), which, considering the nomadic nature of SR farming in many parts of the region, would greatly improve the understanding of the local epidemiological scenario and PPR transmission routes. 

The success of eradication programmes is heavily dependent on a strong legal framework supporting the work of national veterinary services, especially from stage 3 onwards of the PPR GCES. PPR should be categorised as a notifiable disease, and surveillance and control actions should be programmed and budgeted as part of comprehensive national (NSP) and regional workplans. While a Regional PPR Strategy for Central Asia was in fact developed in 2017, it is yet to be enacted and would also require an update in light of the progress and the challenges that emerged in the last years. In addition, some of the considered countries have yet to endorse a PPR NSP, whose review by the PPR GEP Secretariat would ensure that the national strategies are sound and harmonised among neighbouring countries. Among the most common legislative hindrances encountered in the region are difficulties in imposing certain emergency measures in case of PPR outbreaks and particularly compensating farmers should their animals be culled for eradication purposes.

Another critical factor for the effective implementation of control activities is the engagement of key stakeholders along the value chain, including farmers, traders, policy-makers, public and private veterinarians and veterinary paraprofessionals, as well as wildlife authorities and specialists. Awareness campaigns based on different strategies (including the distribution of informative material, organisation of congresses and meetings) should be conducted to ensure all actors are aware of the clinical manifestations of PPR, its consequences for the SR sector, the rationale behind control measures and the modality of PPR reporting. Relevant stakeholders should also be actively involved in the preparation of the NSP and PPR emergency response plans. These activities are of the utmost importance, not only in areas where PPR is endemic but also in PPR-free countries to ensure the maintenance of their freedom status. A widespread shortcoming identified here is the lack of studies on the socioeconomic impact of PPR in the region or in specific countries, which would help inform control and eradication activities, raise awareness, support resource mobilisation and possibly promote cross-border cooperation.

Ultimately, many of the mentioned shortcomings are at least in part ascribable to funding issues. The budget for PPR-related activities is highly variable from country to country and, in some cases, appears inadequate compared to the respective disease burden and SR population size. Aside from donations of diagnostic materials or vaccine stocks, which countries such as Georgia and Kyrgyzstan previously benefited from, the PPR GEP Secretariat, together with other partners, also provides assistance and training to build capacity for attracting donors, mobilising resources in an effective manner and ultimately turning the PPR NSP into an investment plan. Importantly, resource partners have strongly signalled that countries should first mobilise funds both internally and externally to demonstrate their engagement in PPR eradication activities by preparing their own investment and advocacy plans for seeking funding. Donors will then be more likely to join forces with countries to fill in the funding gaps.

## 6. Conclusions

The epidemiological situation and the status of eradication activities in West Eurasia, Central and Eastern Asia present some common features but also unique challenges compared to other regions targeted by the PPR GEP. The proximity of endemic and free countries means that eradication efforts should focus not just on eliminating PPR from infected areas but also on preventing the spread of the disease to new territories with naïve SR populations. For officially or historically PPR-free countries, the development of national emergency response plans, incorporating all elements of contingency specified in the WOAH dossier questionnaire, should be a priority. In addition, the recent history of PPR outbreaks in susceptible wild ruminants in the considered countries calls for particular attention to disease surveillance in wildlife. The use of molecular epidemiology to help define PPRV transmission pathways between countries and host species should also be a priority and, where possible, involve whole genome phylogenetic analysis. These goals cannot be pursued without a truly coordinated approach, in which countries and institutions cooperate at scientific, technical and policy levels. Implicit in such an approach is the need for transparency and open, improved communication between countries, which we consider must be enhanced in the region to achieve the clear socioeconomic and trade benefits of PPR eradication. Towards this aim, the insights provided in this review will hopefully facilitate the harmonised planning of the next steps towards the eradication of PPR in the region. During the second phase of the PPR GEP, the RAG must reinforce its efforts to play a key role in this harmonisation, fostering a sense of ownership of the PPR eradication strategy and implementation at the regional level and encouraging countries to regularly conduct PMAT self-assessments. The timely acquisition of regional PPR freedom in West Eurasia is achievable, aided by the number of historically free countries and progress made to date, and would represent an exemplary success story to other regions involved in the PPR GCES.

## Figures and Tables

**Figure 1 animals-12-02030-f001:**
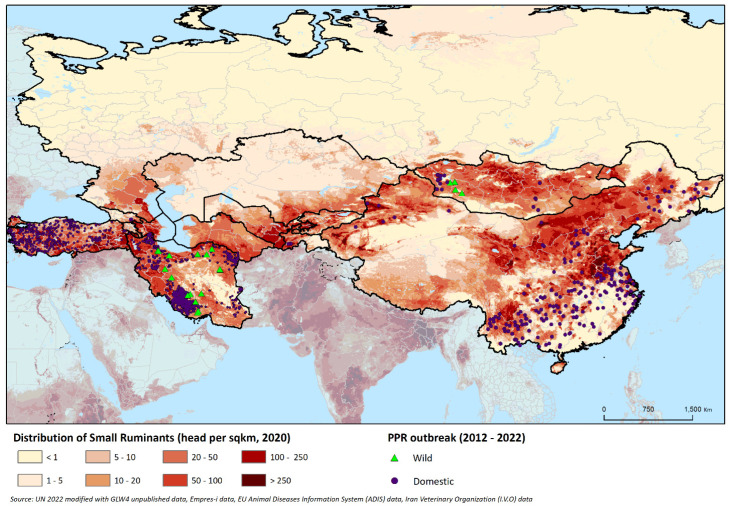
Map of the region showing the distribution of domestic SR (sheep and goats) at the subregional level (adjusted to FAOSTAT population figures for 2020) and the location of reported PPR outbreaks that occurred between 2012 and 2022 in domestic and wild animals. PPR outbreak data were obtained from EMPRES-i [25], from EU-ADIS (for Turkey) [26] and through the Iran Veterinary Organization (for the Islamic Republic of Iran) [27]. The density of domestic SR (head per square km) is indicated by the colour shading, and PPR outbreaks in domestic and wild animals are denoted by purple dots and green triangles, respectively.

**Figure 2 animals-12-02030-f002:**
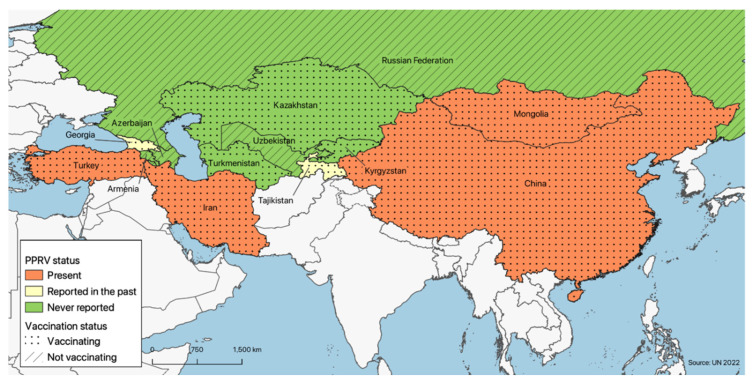
Map of the region under consideration showing each country’s PPR epidemiological and current vaccination status, retrieved respectively from OIE WAHIS and from Roadmap meetings. See main text for further details.

**Figure 3 animals-12-02030-f003:**
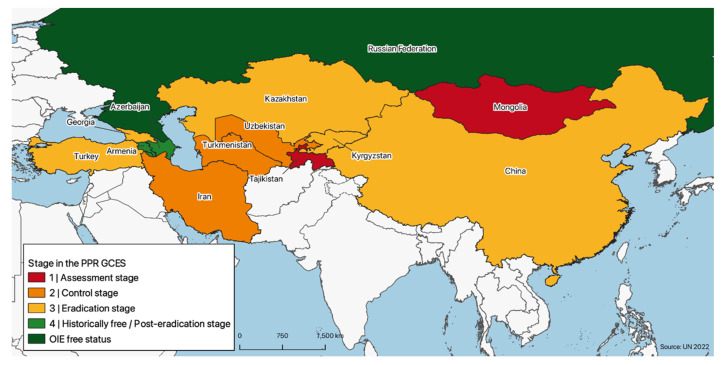
Map of the region detailing each country’s self-reported stage within the stepwise approach of the PPR GCES, updated in October 2021, except for Mongolia (last update in 2017) and Tajikistan (last update in 2019).

**Table 1 animals-12-02030-t001:** Sheep and goat populations at regional and national levels, according to latest available figures (2020) in FAOSTAT.

Country	Sheep	Goats	Total
Armenia	639,598	22,934	662,532
Azerbaijan	7,483,725	605,925	8,089,650
China	173,095,534	133,583,755	306,679,289
Georgia	841,900	49,700	891,600
Iran (Islamic Republic of)	46,587,010	16,663,721	63,250,731
Kazakhstan	17,749,598	2,307,969	20,057,567
Kyrgyzstan	5,508,032	770,704	6,278,736
Mongolia	30,049,428	27,720,253	57,769,681
Russian Federation	20,654,963	1,962,609	22,617,572
Tajikistan	3,818,750	1,950,635	5,769,385
Turkey	42,126,781	11,985,845	54,112,626
Turkmenistan	13,969,559	2,327,988	16,297,547
Uzbekistan	18,829,200	3,629,600	22,458,800
TOTAL	381,354,078	203,581,108	584,935,716

**Table 2 animals-12-02030-t002:** Reported national budgets for activities related to PPR in 2017–2020. Figures originally given in local currency are provided in USD (according to the exchange rate on 1st January of the respective year). Unless other donors are specified in the footnotes, all funds came from government sources. N/A indicates not reported.

Country	2017	2018	2019	2020
Armenia	USD 2 million	USD 2.7 million	USD 2.9 million ^1^	USD 3 million ^1^
Azerbaijan	N/A	N/A	N/A	N/A
China	USD 23 million	USD 24.6 million	USD 23.2 million	USD 23 million
Georgia	USD 2.7 million	USD 2.8 million	USD 3 million	USD 3.2 million
Iran (Islamic Republic of)	N/A	N/A	N/A	N/A
Kazakhstan	USD 32,674	USD 25,001	USD 82,189	USD 375,613
Kyrgyzstan	USD 1.8 million	USD 2.9 million	USD 2.7 million	USD 2.7 million
Mongolia	N/A	N/A	N/A	N/A
Russian Federation	N/A	N/A	N/A	N/A
Tajikistan	N/A	N/A	N/A	N/A
Turkey	USD 14.2 million	USD 15.8 million	USD 8.6 million	USD 8.5 million
Turkmenistan	USD 1243	USD 1240	USD 1238	USD 857
Uzbekistan	USD 18,000	USD 18,500	USD 18,500	USD 20,000

^1^ Including donations equal to USD 120,000 from the USA Defense Threat Reduction Agency (DTRA).

## Data Availability

Not applicable.

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
