# Peer review of "Peste des Petits Ruminants in Central and Eastern Asia/West Eurasia: Epidemiological Situation and Status of Control and Eradication Activities after the First Phase of the PPR Global Eradication Programme (2017–2021)"

_animals, 2022, doi:10.3390/ani12162030_

Round 1

Reviewer 1 Report

This paper provides a review of the implementation of the first 5 years of the PPR Global Eradication Programme (GEP) in 13 contiguous countries belonging to Eastern Europe, Transcaucasia, Central and East Asia. The authors present the status of the stepwise control and eradication process in the 13 countries by comparing data collected through regional Roadmap meetings organized by PPR GEP, national self-assessments using the PPR Monitoring and Assessment Tool (PMAT), regional PPR strategies, the scientific literature, and a series of official databases of disease occurrence and small ruminant population numbers. The information presented will be of significant interest to the scientific community and represents one of the only examples of an effort to synthesize information from the grey literature (e.g. workshop reports and outputs from PMAT) with scientific reports of PPR and global data on disease occurrence and small ruminant populations to examine the status of PPR Global Control and Eradication Strategy (GCES) across a large region spanning Europe and Asia. 

The manuscript is well written and the methodology is appropriate and clearly documented. The authors have incorporated a review of reports of PPR in wild ruminants and indicated where there are gaps in surveillance or consideration of susceptible wildlife species and domestic atypical hosts. The individual country reviews are very useful, demonstrate the heterogeneity in the region when it comes to PPR status, and contribute to the assessment of the opportunity and challenges of moving though the PPR GCES steps at an epidemiologically relevant scale. The introduction mentions the different regions established by PPR GCES for Regional Roadmap meetings and the vision of an “epizone” approach to group countries that are connected epidemiologically and/or are at a similar stage of PPR GCES but this is not expanded on in any detail in the discussion. The figures and tables provided in the manuscript are well designed and complement the material presented in the text. Overall this manuscript represents a significant contribution to examining and improving the global efforts to eradicate PPR and should be published. A few specific comments below:

Line 54 – 55: The authors refer to PPR being responsible for “clinical outbreaks in many wild ruminant species” and cite reference 5 (Dou, 2020). The data does not support clinical outbreaks in many wild species. I suggest a wording change to “evidence of infection” rather than clinical outbreaks and/or indicate “multiple” instead of “many” wild ruminant species.

Line 134: Missing “the” in …..at “the” national level using….

Line 173-178 (Figure 1): Map does not include any “wild” PPR outbreaks noted in China despite reference in the manuscript to PPR in wild ruminants in China documented in the scientific literature. If specific criteria were used to identify reports of PPR in wildlife as outbreaks which excluded the China reports please provide details in the methodology and figure legend. 

Line 443. Reference is made to SR vaccination occurring in Mongolia in “the following weeks” of the initial outbreak first reported in August 2016. As noted further on in the description of PPR in Mongolia the vaccination of SR took place in October 2016. Suggest noting the number of weeks after the initial outbreak SR vaccination was initiated if known. 

Line 516: Suggest adding “the” to sentence …….imminent threat that “the” Thrace region poses…… 

Line 634-635: Suggest adapting the final sentence to read “Donors will then be more likely to join forces with countries to fill in the funding gaps” unless there is assurance from donors that they will fill funding gaps.  

Author Response

  • Line 54 – 55: The authors refer to PPR being responsible for “clinical outbreaks in many wild ruminant species” and cite reference 5 (Dou, 2020). The data does not support clinical outbreaks in many wild species. I suggest a wording change to “evidence of infection” rather than clinical outbreaks and/or indicate “multiple” instead of “many” wild ruminant species.
    • To retain the original focus on PPR pathogenicity rather than the capacity to cause infection in wild species, we opted to replace reference 5 with several references documenting clinical PPR outbreaks in wildlife. All the subsequent references were numbered accordingly. The word “many” was also changed to “multiple” as suggested by the reviewer.
  • Line 134: Missing “the” in …..at “the” national level using….
    • Done as suggested.
  • Line 173-178 (Figure 1): Map does not include any “wild” PPR outbreaks noted in China despite reference in the manuscript to PPR in wild ruminants in China documented in the scientific literature. If specific criteria were used to identify reports of PPR in wildlife as outbreaks which excluded the China reports please provide details in the methodology and figure legend. 
    • The lack of Chinese PPR outbreaks in wild animals in Figure 1 is not due to the exclusion of the respective reports, but to their absence. Unfortunately, all the Chinese outbreak events with known date and location reported through EMPRES-i occurred in domestic ruminants. No “wild” PPR outbreaks could be retrieved from OIE WAHIS either. However, we were able to show wild ruminant outbreaks in the case of Iran since these data were in the reports from the Iran Veterinary Organisation. To reflect this the word ‘reported’ has been added (line 175) and information on the methodology for sourcing the outbreak data, which was already in the methods section, has also been added to the figure legend (lines 176-178).
  • Line 443. Reference is made to SR vaccination occurring in Mongolia in “the following weeks” of the initial outbreak first reported in August 2016. As noted further on in the description of PPR in Mongolia the vaccination of SR took place in October 2016. Suggest noting the number of weeks after the initial outbreak SR vaccination was initiated if known. 
    • According to the data presented by Mongolian representatives during the Roadmap meetings, the vaccination campaign lasted from 10 to 30 October 2016. The information has been added at line 444-5.
  • Line 516: Suggest adding “the” to sentence …….imminent threat that “the” Thrace region poses……
    • Done as suggested. 
  • Line 634-635: Suggest adapting the final sentence to read “Donors will then be more likely to join forces with countries to fill in the funding gaps” unless there is assurance from donors that they will fill funding gaps.  
    • Done as suggested.

Reviewer 2 Report

Legnardi and colleagues present a very detailed review about the current epidemiology of peste des petits ruminants and the status of its control and eradication in Central and Eastern Asia/West Eurasia.

PPR is a transboundary and highly infectious notifiable disease of small ruminants, endemic in various countries of Africa and Asia. The FAO and OIE set the goal for global PPR eradication by 2030. This review summarized the first stage of the eradication program.

The review is very detailed, informative and one can feel that a lot of information was brought together. It is very well written and the English does not require any further editing. Therefore, there is not much to suggest for revision.

The author stated that a lot of information was obtained from a regional Roadmap meeting of the participating countries otherwise literature references are given. However, these passages are not cited. Some passages do not have a reference at all and it’s not clear where the respective information is coming from. This is the only thing I have to remark.

Author Response

  • As correctly stated, a major part of the data included in the present review has been extrapolated from presentations delivered by country representatives during Roadmap meetings, workshops and other activities conducted in the region of interest. We believe that this manuscript offers a rare occasion to present this body of information in a publishable and harmonized form. Nonetheless, we also acknowledge that the absence of references may hinder the navigation of some sections. For this reason, some passages (lines 258-9, 296, 352, 469, and 534-6) were edited to clarify the source of the respective information. Additional references (either newly inserted or already cited in the paper) have also been inserted at lines 159, 310, 369, 401, 418, 451, 504, and 519 to support some data which could be retrieved not only from the grey literature, but also from published works.
